# Detecting Nonlinear Interactions in Complex Systems: Application in Financial Markets

**DOI:** 10.3390/e25020370

**Published:** 2023-02-17

**Authors:** Akylas Fotiadis, Ioannis Vlachos, Dimitris Kugiumtzis

**Affiliations:** 1Department of Electrical and Computer Engineering, Aristotle University of Thessaloniki, 54124 Thessaloniki, Greece; 21st Department of Neurology, Medical School, Aristotle University of Thessaloniki, 54124 Thessaloniki, Greece

**Keywords:** nonlinear interactions, structural break, Granger causality, causality networks, financial crisis

## Abstract

Emerging or diminishing nonlinear interactions in the evolution of a complex system may signal a possible structural change in its underlying
mechanism. This type of structural break may exist in many applications, such as in climate and finance, and standard methods for change-point
detection may not be sensitive to it. In this article, we present a novel scheme for detecting structural breaks through the occurrence or vanishing of nonlinear causal relationships in a complex system. A significance resampling test was developed for the null hypothesis (H_0_) of no nonlinear causal
relationships using (a) an appropriate Gaussian instantaneous transform and vector autoregressive (VAR) process to generate the resampled multivariate time series consistent with H_0_; (b) the modelfree Granger causality measure of partial mutual information from mixed embedding (PMIME) to estimate all causal relationships; and (c) a characteristic of the network formed by PMIME as test statistic. The significance test was applied to sliding windows on the observed multivariate time series, and the change from rejection to no-rejection of H_0_, or the opposite, signaled a non-trivial change of the underlying dynamics of the observed complex system. Different network indices that capture different characteristics of the PMIME networks were used as test statistics. The test was evaluated on multiple synthetic complex and chaotic systems, as well as on linear and nonlinear stochastic systems, demonstrating that the proposed methodology is capable of detecting nonlinear causality. Furthermore, the scheme was applied to different records of financial indices regarding the global financial crisis of 2008, the two commodity crises of 2014 and 2020, the Brexit referendum of 2016, and the outbreak of COVID-19, accurately identifying the structural breaks at the identified times.

## 1. Introduction

One of the most important assumptions in time series analysis is the notion of stationarity. Under this assumption, all the statistical properties of a time series do not change over time. The majority of the analytical tools and statistical models rely on this assumption, and any violation may lead to inaccurate conclusions. However, when dealing with real data, the assumption of stationarity is often violated, and one interesting type of violation is the change in the underlying dynamics. This is referred to as structural change or break and has been in the focus of studies on real complex systems, such as financial markets [1,2,3], brain dynamics and connectivity [4,5,6,7], and climate [8,9]. Some studies have investigated multiple structural breaks in a time series record [1,10,11,12].

In a statistical setting, a structural break is examined as a change point that violates the stationarity property of the time series or signals the shift in a non-stationarity property, e.g., in the form of mean jumps, trends, or other morphological characteristics of the time series. Change point detection has been vastly studied in the statistical literature, both in univariate and multivariate time series, using parametric statistical tests to detect changes in the first- and second-order moments (mean, variance, autocovariance) [13]. Non-parametric methods, including CUSUM, bootstrap techniques, and kernel estimators, have also been used [14,15]. The methods originally proposed for off-line change-point detection have also been adapted for online detection [16,17].

While statistical moments estimated on univariate and multivariate time series have also been used to detect regime changes in real-world complex systems, other more advanced measures that could detect nonlinear effects have been found to be more appropriate, e.g., in the detection and prediction of epileptic seizures [18,19] and the dynamics in economy and finance [20,21]. In recent years, interdependence measures on multivariate time series have been applied to estimate the connectivity structure of complex systems [22]. These measures are broadly referred to as Granger causality measures [23,24], and they have been used widely in neuroscience [25,26], climate studies [27], and more recently, in finance [28,29,30,31], where synergistic effects have also been investigated [32]. Figure 1 summarizes the main categories of the methods used for change-point detection in univariate and multivariate time series.

In this article, we reported on a novel approach for tracking structural changes through the detection of nonlinear interactions present in the underlying mechanisms of financial systems. The main assumption is that before a structural change event in a complex system, nonlinear causalities could occur or vanish, or existing nonlinear causalities could be significantly stronger or weaker [33,34,35]. To estimate the presence of nonlinear causal effects in a high-dimensional system, such as the financial markets, we relied on the measure of partial mutual information from mixed embedding (PMIME) [36,37], which detects the direct causal effects also present in high-dimensional systems [24,38] and in different applications [29,39,40,41,42]. The PMIME measure was used to derive the test statistic for the null hypothesis of the trivial (in the sense of none or linear) causal structure of the underlying complex system, where the alternative hypothesis concerned the presence of nonlinear causal relationships. Since our interest was in the overall causality structure of the system, different global characteristics of the estimated causality network from PMIME were used as test statistics. Due to the lack of a parametric asymptotic null distribution of the test statistics, a randomization test was developed to generate the randomized multivariate time series from a linear vector autoregressive model that was fitted to the original multivariate time series.

The rationale of the proposed approach is that the estimated causality network with PMIME, in the presence of significant nonlinear causal relationships between the observed variables of the complex system, is different from the respective network when the nonlinear causal relationships are significantly weaker or absent. The focus of the study was on detecting such changes in data records of complex systems evolving over long time periods, especially financial markets. For this, the test was applied repeatedly according to rolling windows of the multivariate time series record. First, we confirmed the appropriateness of the suggested approach to the simulated data and then we applied the computational setup to different financial time series records.

The structure of the paper is as follows. In Section 2, the Granger causality measure PMIME, the network indices used as test statistics, and the generation of the randomized time series for the test are presented. In addition, the simulation setup and data are described. In Section 3, the results of the computations on simulated and financial data are presented and discussed. Finally, in Section 4, the conclusions are drawn.

## 2. Methods and Data

Next, the overall approach for detecting nonlinear interactions and structural change is presented and then the different parts of the approach are discussed in detail. More specifically, we first present the approach for detecting nonlinear interactions at each time window, and after, we present the procedure across the time windows.

### 2.1. The Approach for Detecting Nonlinear Interactions and Structural Change

Let $\left\{x_{1,t},x_{2,t},\ldots ,x_{K,t}\right\}$, and t=1,…,N denote a multivariate time series of *K* variables $X_{1},\ldots ,X_{K}$ and length *N*, split in time windows of length *n*, overlapping or non-overlapping. The objective was to detect changes in the nonlinear causality structure of the underlying system across the time windows. Thus, first the nonlinear causality structure at each time window was estimated and tested for significance. For the estimation of nonlinear causality between all pairs of the *K* variables, the partial mutual information from mixed embedding (PMIME) was used, presented briefly in Section 2.2. The causality network was formed with the *K* variables as nodes and the PMIME values for all ordered pairs as connections, and different characteristics of the network were quantified by network indices, as discussed in Section 2.3. Any of these network indices were used as test statistics for the significance test for the nonlinear causality structure of the system of the *K* observed variables or subsystems. The null hypothesis ($H_{0}$) was that the causality structure was trivial and there were no nonlinear causal relationships among the *K* variables. As the test statistic (the network index) has no parametric asymptotic null distribution, a resampling test was performed. For this, the *M* surrogate (randomized) multivariate time series of the *K* variables and length *n* were generated, so that they had the same marginal distribution and the same linear auto- and cross-correlation as the original time series while lacking any nonlinear causality possibly existing in the original time series. To achieve this, each of the *K* original univariate time series was statically transformed to have a Gaussian marginal distribution, a vector autoregressive (VAR) model was fitted to these *K* Gaussian time series, and the VAR model was used as the data-generating process to obtain the *M* Gaussian time series, which were then statically transformed back to their original marginal distribution. The generation of the surrogate time series is presented in detail in Section 2.4. The PMIME was computed on each of the *M* surrogate multivariate time series, the *M* causality networks were obtained, and subsequently, the *M* values of the selected network index were derived. These were the *M* values of the test statistic forming the empirical null distribution and the significance bounds for the test statistic (the network index). From this, the *p*-value of the test and the test decision at a significance level α could be derived. The surrogate test for nonlinear causality is illustrated in Figure 2.

We repeated the test for each of the sliding windows of size *n* across the time series record of length *N* (n≪N), and in this way, we obtained the profile of the network index, together with its significance bounds for the whole recording period. Then, a structural break in the complex system of the type of emergence or vanishing of nonlinear causality structure could be detected by a change in the significance of the network index. An illustrative example is provided in Figure 3, where the time series record was split to seven non-overlapping windows and the test for nonlinear causality by a specific network index was applied to each of the seven windows.

It was observed that the network index was within the significance bounds in the first three time windows and then drops out of the significance bounds in the next four time windows, indicating that in the fourth time window, there was a structural break. Quantitatively, the change was detected in terms of the *p*-values of the tests at each time window, as compared to the significance level α. Then, the type of the change, emergence or vanishing, of the nonlinear causality structure was further identified by the specific network index used as test statistic.

### 2.2. The Granger Causality Measure PMIME

The Granger causality measure PMIME is an information-based, parameter-free measure defined in terms of mutual information (MI) and conditional MI (CMI) [36,37]. The MI and CMI are estimated using the *k*-nearest-neighbor estimator, as it is more stable and more efficient than the others [36,43], and it allows the estimation of entropies on higher dimensional vector variables. The latter is particularly relevant for the estimation of direct causality on a time series of high dimension *K* [37].

The PMIME is considered a modification of the partial transfer entropy using dimensional reduction, as explained below. First, recall that the transfer entropy (TE) measures the causality from a driving variable $X_{1}$ to a response variable $X_{2}$ in a bivariate time series $\left\{x_{1,t},x_{2,t}\right\}$, t=1,…,n. TE is defined by the CMI, as follows:$${TE}_{X_{1}\to X_{2}}=I\left(x_{2,t+1};x_{1,t}|x_{2,t}\right),$$
where the embedding vector $x_{1,t}$ contains the information of $X_{1}$ from present to past, simply defined up to a maximum lag *L*, $x_{1,t}=\left[x_{1,t},x_{1,t-\tau },\ldots ,x_{1,t-(L-1)\tau }\right]$ (more generally, a time lag τ between the successive components would be used, but for a time series regarding discrete time systems, such as in finance, typically τ=1). The same notation stands for $x_{2,t}$, using typically the same maximum lag *L*. In essence, TE measures the information in the present and the past of $X_{1}$, explaining the future of $X_{2}$ that is not contained already in the present and past of $X_{2}$.

The partial transfer entropy (PTE) extends the TE in the presence of other observed variables. Thus, for the causality relationship $X_{1}\to X_{2}$ in the presence of the other K−2 variables stacked in $Z=[X_{3},\ldots ,X_{K}]$, PTE is defined similarly, but the conditioning term included, on top of the embedding vector $x_{2,t}$, the K−2 respective embedding vectors of the other K−2 variables stacked in $z_{t}$, with a total of (K−2)L components. In this way, PTE measures the direct causality of $X_{1}$ to $X_{2}$, defined as follows:$${PTE}_{X_{1}\to X_{2}}=I\left(x_{2,t+1};x_{1,t}|x_{2,t},z_{t}\right),$$
quantifying the information in the present and past of $X_{1}$ while explaining the future of $X_{2}$ that is not contained already in the present and past of any of the other variables ($X_{2}$ and the other K−2 variables).

While PTE is conceptually suitably defined to measure direct causality, it is not practically useful as the estimation of the CMI for its definition failure when either *K* or *L* (or both) is relatively large. The estimation of CMI relies on the estimation of entropy terms, and for PTE, the entropy term of the vector variable has all arguments in the CMI regarding the highest dimension of KL+1, as components. Even for simple scenarios with relatively low *L* and *K*, the estimation of the entropy terms is inaccurate unless the time series length *n* is long. Especially, with the setting of sliding windows, *n* should remain short to be able to identify the structural break accurately (and early).

The PMIME addresses the problem of dimensionality due to the use of the *K* embedding vectors, each of *L* components, where the latter are referred to as lag variables [36,37]. Specifically, PMIME progressively builds the so-called mixed embedding vector $w_{t}$ that contains the most informative lag variables for the future of the response variable $X_{2}$ ($x_{2,t+1}$), regarding a small subset of the set of all KL lagged variables. It begins with an empty mixed embedding vector $w_{t}^{0}$ (at step j=0), at each step j+1. First, the lag variable $w^{*}$ is found to have the largest amount of information for $x_{2,t+1}$ that is not already included in the components of the current mixed embedding vector $w_{t}^{j}$, maximizing the CMI $I(x_{2,t+1};w^{*}|w_{t}^{j})$. Then, the significance of $I(x_{2,t+1};w^{*}|w_{t}^{j})$ is tested using time-shifted surrogates [44], and if not found significant, the procedure terminates and $w_{t}=w_{t}^{j}$. Otherwise, $w_{t}^{j+1}=\left[w_{t}^{j},w^{*}\right]$ and repeats the same step, increasing *j* by one. The mixed embedding vector $w_{t}$ could contain lag variables of the driver $X_{1}$, the response $X_{2}$, and the rest of the variables in *Z*, denoted by $w_{t}^{X_{1}}$, $w_{t}^{X_{2}}$, and $w_{t}^{Z}$, respectively. Thus, the predictive information of the response $X_{2}$, solely from the driving $X_{1}$, is quantified by $I(x_{2,t+1};w_{t}^{X_{1}}|w_{t}^{X_{2}},w_{t}^{Z})$ and normalized by the mutual information of $x_{2,t+1}$ and $w_{t}$, so that the PMIME causality measure is defined, as follows:(1)$$PMIME_{X_{1}\to X_{2}}=\frac{I(x_{2,t+1};w_{t}^{X_{1}}|w_{t}^{X_{2}},w_{t}^{Z})}{I(x_{2,t+1};w_{t})}.$$
If there is no causal effect from $X_{1}$ to $X_{2}$, then $w_{t}^{X_{1}}$ is empty, and PMIME is zero; otherwise, PMIME is positive and obtains its maximum value one when the only predictive information on $x_{2,t+1}$ is from $X_{1}$ ($w_{t}^{X_{2}}$ and $w_{t}^{Z}$ were empty).

The PMIME is computed for each directed pair of the *K* observed variables yielding the weight matrix or adjacency matrix, if the positive values are set to one, and subsequently, the corresponding causality network of weighted or binary connections, respectively.

### 2.3. Network Indices

As test statistics for the test for nonlinear causality, we used different network indices that are believed to capture different characteristics of the network structure. The indices were computed on the causality networks formed by PMIME with binary connections (a connection existed if PMIME >0) and weighted connections (the weight was the PMIME value).

The network indices of interest concerned the first and second moments of the degree *k* and strength *s* distribution. For binary connections, these were the mean ave(k) and the standard deviation SD(k) of the degree *k* of each node, and for weighted connections, ave(s) and SD(s) of the strength *s* of each node, where degree or strength of a node is the sum of the incoming and outgoing binary or weighted connections, respectively. The rationale of using the weighted connections was to focus not only on the existence of a causality relationship but also on its strength. For the PMIME, the presence of lag terms of the driving variable in the mixed embedding vector indicated the existence of causal effect of the driving variable on the response variable, but the contribution of the driving lag variables in explaining the response could be small or large, and it was quantified by the PMIME value, i.e., the weighted connection. Under the same rationale and with the strong sparsity of the networks, we also defined the mean and standard deviation of only the positive weighted connections, denoted as $ave(s^{+})$ and $SD(s^{+})$, respectively.

### 2.4. Surrogate Data Generation

Here, we present the generation of the randomized multivariate time series consistent with $H_{0}$, indicating that the system generating the *K* time series had no nonlinear interactions (only linear causal relationships among the *K* variables). In particular, according to the $H_{0}$, there were no nonlinear autocorrelations and cross-correlations in the observed multivariate time series. Instead of generating surrogate data matching directly to the linear autocorrelations and cross-correlations, together with the marginal distribution of each of the *K* time series, as carried out in constrained realization approaches (e.g., [45]), we used the typical realizations of a fitted vector autoregressive (VAR) model to the *K*-variate time series. Before the fit of a VAR model, a monotone transform was applied to each variable, such as *X*, of the *K* variables in order to have a Gaussian marginal distribution.
(2)$$z=\Phi^{-1}\left(F_{X}\left(x\right)\right),$$
where the obtained variable *Z* has the standard Gaussian distribution with the cumulative density function denoted as Φ(z), and $F_{X}\left(x\right)$ denotes the sample cumulative density function of *X*, given by the naive estimate of rank ordering ($F_{X}\left(x\right)=r/n$, where *r* is the rank of *x* in the list of the *n* observations of *X*). Then, the VAR model of order *P* was fitted to the *K* Gaussianized time series. We set P=L to account for the same maximum lag used in the PMIME and assure that the VAR model captures the linear autocorrelation and cross-correlation up to lag *L*. The VAR model was used as the data generating process, and we obtained a large number *M* of surrogate *K*-dimensional time series, which had the same linear autocorrelations and cross-correlations as the original Gaussianized *K*-dimensional time series. Further, we applied the inverse transform of Equation (Equation 2) to acquire the original marginal distribution
(3)$$x=F_{X}^{-1}\left(\Phi \left(z\right)\right).$$
Under this transform, the surrogate time series preserved the linear autocorrelation, cross-correlation, and thus, the causal relationships among the *K* variables, and also the marginal distribution of each of the *K* variables. We observed at this point that the surrogate data preserved exactly the marginal distribution (the same data points in the original and surrogate time series) and, approximately, the linear relationships. In particular, the *K* time series, derived by the transform in Equation (Equation 2), were only Gaussian in the marginal distribution, and the underlying *K*-dimensional process could still be non-Gaussian (with Gaussian marginals). Indeed, it would be non-Gaussian if the $H_{0}$ was not true. However, the data generating process, the fitted VAR model, was a Gaussian process, so that after the inverse transform in Equation (Equation 3), the linear correlations may not match these of the original time series. This has been reported for the test for nonlinearity on scalar time series, and it held true for the multivariate case [46,47,48]. Nonetheless, in our setting, we did not focus on the match of the linear correlation structure but, rather, the overall causality structure as quantified by a network index. The rationale for this test was that the PMIME causality structure of the original and surrogate multivariate time series was the same if the original multivariate time series did not include nonlinear causality, whereas the presence of nonlinear causality in the original multivariate time series resulted in a different causality structure, as estimated by PMIME, that could not be preserved by the surrogate time series.

### 2.5. Structural Break Detection Algorithm

Having presented the different parts of the test for nonlinear causality, we discuss here how the test was applied to detect structural change in the complex system generating the multivariate time series. As shown in Figure 3, a long time series of length *N* was split into overlapping or non-overlapping windows of size *n*. For each time series of length *n*, we applied the PMIME and the test for nonlinear causality. In particular, the *p*-value of the test was derived by the rank ordering of the original test statistic (network index) in the list of the M+1 statistics (using the correction for the empirical cumulative function in [49], we computed the *p*-value for the one-sided test as $1-\left(r^{0}-0.326\right)/\left(M+1+0.348\right)$, where $r^{0}$ is the rank of the original statistic value in the ordered list of M+1 values). Rejection of the $H_{0}$ was then established if p<α. In the simulations, we used α=0.05, but if dependent on the data setting, other values of α could be used. For example, in the case of a system that always exhibited nonlinear causality and the structural change regarded a significant change in the nonlinear causality, the tests would reject all sliding windows for α=0.05 but would detect the change in the nonlinear causality if a smaller α was used. Therefore, in the simulations, we showed the profile of *p*-values.

We considered that the structural change could go both directions, i.e., the nonlinear causality structure could emerge or vanish. Thus, a structural change was detected if, for two consecutive windows, there was no rejection, and the third window resulting in a rejection (nonlinear causality structure emerged). This was also the case if for two consecutive windows, there were rejections, and the third window offered no rejection (nonlinear causality structure vanished). The procedure is demonstrated in the Algorithm 1.
**Algorithm 1** Structural break detection1:Split multivariate time series into *p* windows of size *n*2:Test for nonlinear causality in window 1 and get boolean test decision Reject(1).3:Test for nonlinear causality in window 2 and get boolean test decision Reject(2).4:**for**i←3,p **do**5:    Test for nonlinear causality in window i and get boolean test decision Reject(i)6:    **if** (NOT Reject(i−2)) AND (NOT Reject(i−1)) AND Reject(*i*) **then**7:        FLAG structural break type 1       % nonlinear causality structure emerges8:    **else if** Reject(i−2) AND Reject(i−1) AND (NOT Reject(*i*)) **then**9:        FLAG structural break type 2       % nonlinear causality structure vanishes10:    **else**11:        continue12:    **end if**13:**end for**

The performance of the proposed procedure was evaluated in simulated systems and different financial systems.

### 2.6. Data from Synthetic and Financial Systems

The multivariate time series used in our simulation study were generated from multiple known chaotic dynamical systems, such as the so-called coupled Hénon maps and the causal Hénon maps, the causal logistic map, as well as stochastic systems, a nonlinear VAR of order 4, NLVAR(4), and a VAR(4). Only discrete-time systems were considered, as the financial time series sets at the center of the study were in discrete time. The dimension of the system of causal logistic maps was K=5 (5 coupled logistic maps), and for the other systems, it was K=10. The length of each time series or time window was n=1024. The systems are briefly presented below.

#### 2.6.1. Coupled Hénon System

The system of *K* coupled Hénon maps (CoupledHM) was defined as [37]
$$\;\begin{array}{cc} x_{i,t}=1.4-x_{i,t-1}^{2}+0.3x_{i,t-2}, & for\;\;i=1,K\\ x_{i,t}=1.4-{\left(0.5C\left(x_{i-1,t-1}+x_{i+1,t-1}\right)+\left(1-C\right)x_{i,t-1}\right)}^{2}+0.3x_{i,t-2}, & for\;\;i=2,\ldots ,K-1, \end{array}$$
where the coupling strength is the same for all couplings and set to C=0.3 regarding weak coupling before the synchronization limit. The network structure is an open ring, as shown in Figure 4, along with a realization of length n=1024 and K=10.

#### 2.6.2. Causal Hénon System

The second system of causal Hénon maps (CausalHM) regarded also the coupling of Hénon maps, but with a different coupling structure, where a variable of a Hénon map drove the next Hénon map, and it was defined as [50]
$$\;\begin{array}{cc} x_{i,t}=1.4-x_{i,t-1}^{2}+0.3x_{i,t-2}, & for\;\;i=1\\ x_{i,t}=1.4-C\left(x_{i-1,t-1}x_{i,t-1}\right)+\left(1-C\right)x_{i,t-1}^{2}+0.3x_{i,t-2}, & for\;\;i=2,\ldots ,K, \end{array}$$
where again the coupling strength is fixed at C=0.3. This system has a similar complexity to the system CoupledHM, but a different coupling structure (unidirected causality). The network structure and a realization of length n=1024 and K=10 are shown in Figure 5.

#### 2.6.3. Causal Logistic Map

The system of causal logistic maps (CausalLM) has a similar coupling structure as the system CausalHM, defined as
$$\begin{array}{cc} x_{1,t}=x_{1,t-1}\left(4-4x_{1,t-1}\right), & \\ x_{i,t}=x_{i,t-1}\left(4-4x_{i,t-1}-Cx_{i-1,t-1}^{2}-e_{t}\right),\;x_{i,t}=mod\left(x_{i,t},1\right) & for\;\;i=2,\ldots ,K, \end{array}$$
where the coupling strength is fixed at C=0.3. The logistic map is one dimensional, whereas the Hénon map is two dimensional, but it is more complex (larger maximum Lyapunov exponent) and has zero autocorrelation at any lag. It was included in the simulation study to examine the degree to which a linear stochastic system (the fitted VAR) could compensate for the nonlinear causality effects of a purely nonlinear dynamical system. The network structure and a realization of length n=1024 and K=5 are shown in Figure 6.

#### 2.6.4. NLVAR(4) System

This system is a stochastic, nonlinear VAR (NLVAR) process of order P=4 on K=10 variables, expressed as
$$\begin{array}{cc} x_{1,t} & =0.49x_{1,t-2}+e_{1,t}\\ x_{2,t} & =0.49x_{2,t-2}+0.29x_{4,t-1}^{2}-0.31x_{8,t-2}x_{10,t-4}+e_{2,t}\\ x_{3,t} & =0.49x_{3,t-2}+0.29x_{5,t-1}x_{1,t-4}+e_{3,t}\\ x_{4,t} & =0.49x_{4,t-2}+e_{4,t}\\ x_{5,t} & =0.49x_{5,t-2}+e_{5,t}\\ x_{6,t} & =0.49x_{6,t-2}-0.35x_{3,t-2}+0.31x_{9,t-1}^{2}+e_{6,t}\\ x_{7,t} & =0.49x_{7,t-2}+e_{7,t}\\ x_{8,t} & =0.49x_{8,t-2}+0.1x_{7,t-1}^{2}+e_{8,t}\\ x_{9,t} & =0.49x_{9,t-2}+e_{9,t}\\ x_{10,t} & =0.49x_{10,t-2}+0.32x_{9,t-1}+e_{10,t} \end{array}$$
where $e_{i,t}$, i=1,…,10 is the input Gaussian and uncorrelated white noise. The coupling structure is rather random and sparse, as can be seen in Figure 7, together with a realization of length n=1024.

#### 2.6.5. VAR(4) System

We derived a linear VAR process of order P=4 on K=10 variables from the NLVAR(4) process by converting the nonlinear terms to linear ones: dropping the square in a square variable term and removing one lag variable at random from a product term. In this way, the causal relationships $X_{10}\to X_{2}$ and $X_{4}\to X_{3}$ were removed. The system equations are
$$\begin{array}{cc} x_{1,t} & =0.49x_{1,t-2}+e_{1,t}\\ x_{2,t} & =0.49x_{2,t-2}+0.29x_{4,t-1}-0.31x_{8,t-2}+e_{2,t}\\ x_{3,t} & =0.49x_{3,t-2}+0.29x_{5,t-1}+e_{3,t}\\ x_{4,t} & =0.49x_{4,t-2}+e_{4,t}\\ x_{5,t} & =0.49x_{5,t-2}+e_{5,t}\\ x_{6,t} & =0.49x_{6,t-2}-0.35x_{3,t-2}+0.31x_{9,t-1}+e_{6,t}\\ x_{7,t} & =0.49x_{7,t-2}+e_{7,t}\\ x_{8,t} & =0.49x_{8,t-2}+0.1x_{7,t-1}+e_{8,t}\\ x_{9,t} & =0.49x_{9,t-2}+e_{9,t}\\ x_{10,t} & =0.49x_{10,t-2}+0.32x_{9,t-1}+e_{10,t} \end{array}$$
The coupling structure and a realization of length n=1024 of VAR(4) process are shown in Figure 8.

We note that in both NLVAR(4) and VAR(4), the couplings are very weak and may not be detected by PMIME. We could not find a solution for this, as when we increased the coefficients to increase the couplings’ strength, the systems became unstable.

#### 2.6.6. Real Data

The empirical analysis encompassed various financial events that occurred in the last fifteen years, namely the global financial crisis of 2007–2008, the crises in the commodities market in the second half of 2014 and in April of 2020, the Brexit referendum in June 2016, and the COVID-19 pandemic, which began in December 2019. For each event, we used financial assets assumed to be relevant to or affected by the ensuing event. Therefore, for the financial crisis of 2007–2008, we used 37 stocks from the U.S. stock exchange market, covering the period from 2004 until the middle of 2012. For the commodities crisis of 2014, we used the future prices of 8 commodities and Morgan Stanley Capital Indices (MSCI) of 7 countries for the period of the middle of 2012 until 2022. For the Brexit referendum, we used 17 stocks from the FTSE index from the middle of 2012 until the end of 2019. For the COVID-19 pandemic, we used 17 futures from government bonds and indices (from September of 2019 until October of 2020). (All but the MSCI data were obtained from https://finance.yahoo.com (accessed on 14 February 2023), and the MSCI data was sourced from http://www.msci.com, accessed on 1 November 2021). For the first three events, we used the daily log-returns of the close values with a sliding window of size n=300 and step s=150, while for the pandemic and the futures, we used hourly log-returns of the close values with a sliding window of size n=1000 and step s=500.

## 3. Results

In this section, we present the results of the application of the procedure on the simulated and real data. Specifically, in Section 3.1, we demonstrate the structural change detection on a particular setting using simulated data, and then in Section 3.2, we report the results of the nonlinear causality test on the different simulated systems, and in Section 3.3, we present the results on structural break detection in the four financial data records.

### 3.1. Structural Change Detection in Simulated Data

We began with a synthetic example to demonstrate the procedure of detecting the structural change caused by a change in the nonlinear causality structure of the observed complex system. The time series record was a multivariate time series of K=10 variables and length N=3600, and we split it into 12 non-overlapping windows of size n=300, as shown in Figure 9a.

In this example, we designed a gradual transition from a linear system, the VAR(4) system in Section 2.6.5, to a nonlinear system, the causal Hénon maps (CausalHM) in Section 2.6.1. To realize this setting, at each time window, the observed time series is the weighted average of the time series as generated by the two systems, and the weights are complementary percentages, denoted as linear% and nonlinear%, as shown in Figure 9b. We expected that the structural change would occur first at window 7, where the nonlinear part first was larger than the linear part.

At each of the 12 time windows, we computed the PMIME on the original data and the M=100 VAR surrogate data, derived the M+1 causal networks, and computed the network indices on these networks as different test statistics for the test for nonlinear causality. We then computed the *p*-value for each network index at each time window and derived the *p*-value profiles across the 12 time windows, as shown in Figure 10. The variable *L* was set equal to 5.

All network indices tended to provide lower *p*-values on the second half of the time period, where the underlying dynamical system became increasingly nonlinear. The change was best detected by the first and second moment of network strength, restricted only to positive weight connections, $ave(s^{+})$ and $SD(s^{+})$, respectively, as the *p*-value was high for the first part of the time series record and below α in the second part (marginally over α at windows 10 and 11 for $SD(s^{+})$), pointing to the structural change at window 7. The first and second moment of network strength, ave(s) and SD(s), respectively, provided similar *p*-value profiles, but with the drop of the *p*-value below α at window 8 for SD(s) and with an incorrect drop of the *p*-value below α at window 5 for ave(s). The latter was also observed for the mean degree, ave(k), but this test statistic seemed not to be able to differentiate the presence of nonlinear causality with statistical significance, as the *p*-value was lower in the second part but still over α. The SD of the degree, SD(k), performed better as it captured the change from a high to a low *p*-value in the central time windows but also had p>α for windows after 9. Thus, overall, the strength-based statistics performed better than the degree-based statistics, indicating that the intensity of the estimated causal effect was more important than the existence of the causal effect.

### 3.2. Nonlinearities Detection in Simulated Systems

We after focused on the nonlinear causality test and assessed its performance on the simulated systems presented in Section 2.6. We set L=5 for all systems, which was sufficiently large, and, in most cases, much larger than the system lag order to make the proposed procedure essentially parameter-free.

#### 3.2.1. VAR(4) Model

The VAR(4) system is linear (see Section 2.6.5), and we expected the VAR surrogates to successfully capture the original (linear) causality structure. The test results with the different network indices as test statistics for a single realization are shown in Figure 11.

For any network index, the original test statistic value was found well into the empirical null distribution formed by the M=100 surrogate values, yielding a large *p*-value and no rejection of the $H_{0}$ of linear causality structure (lowest *p*-value was 0.08 for the statistic $SD(s^{+})$).

In Figure 12, the distribution of the *p*-values that occurred after conducting 100 simulations is presented. The distribution was relatively uniform, and no systematic clustering of the *p*-values, in particular, at a low level of *p*, was observed for any of the six statistics, indicating no sign of systematic rejection.

However, as shown in Table 1 (first row), the test size was not very good, as the relative frequency of false rejection (Type I error) from 100 realizations was not close to the predefined significance level, a=0.05. The best performing indices were the $SD(s^{+})$ and the $ave(s^{+})$, rejecting the $H_{0}$ for 9 out of 100 realizations, while the network index that performed the worst was the ave(s), which rejected 19 realizations.

#### 3.2.2. NLVAR(4)

For the nonlinear stochastic system NLVAR(4) (see Section 2.6.4), the power of the test was low, as indicated by the relative rejection frequency for the six network indices in Table 1 (second row). The statistic $SD(s^{+})$ had the highest power, rejecting 22 out of 100 trials at α=0.05. This result was mainly attributed to the weak nonlinear interactions in NLVAR(4) that could not be accurately estimated by the PMIME, so the difference between the original time series and the linear surrogates could not be clearly established.

#### 3.2.3. Causal Hénon Map

The power of the test increased for the system of causal Hénon map (CausalHM), as shown in Table 1 (third row). The network indices SD(s) and $SD(s^{+})$ always detected the presence of nonlinear couplings, while the other indices presented lower statistical power. For this system and the particular coupling structure, the standard deviation of degree or strength had larger power than the respective average, as ave(k) and ave(s) presented the lower relative frequency of rejection of $H_{0}$.

#### 3.2.4. Coupled Hénon Map

For the system of the coupled Hénon maps (CoupledHM), as compared to CausalHM, $SD(s^{+})$ had the lowest power while SD(s) had the highest power, as shown in Table 1 (fourth row). However, the level of power for all network indices was lower than for CausalHM, and, overall, the network indices on weighted connections performed better here as well. While CoupledHM is a nonlinear system, we observed that the proposed test did not perform as expected. This could be explained by the fact that the causal connections that this system exhibited could be detected, to some extent, by a linear causality measure. For example, we had estimated the causal network using the conditional Granger causality index CGCI [51], and we had found that it had a sensitivity equal to 0.988. Thus, CGCI almost always detected the existent causalities. Hence, the VAR surrogates captured this structure, and the PMIME measure could not distinguish it from the original one.

#### 3.2.5. Causal Logistic Map

As shown in Table 1 (fifth row), for the system of causal logistic maps (CausalLM), ave(s), $ave(s^{+})$ and SD(s) had the highest statistical power equal to one. On the other hand, the other three indices performed poorly, with the ave(k) index having the lowest power (0.04). Furthermore, in this case, 3 out of 4 weighted indices were able to detect the nonlinear causalities that were present in this chaotic system.

### 3.3. Structural Break Detection in Real Financial Data

The performance of our proposed method for the detection of structural breaks was evaluated on the financial datasets described in Section 2.6.6. For each time window referred to by the end date of the period estimated, we generated 100 VAR surrogates, we estimated the PMIME measure using L=3 and performed the hypothesis test at the significance level α=0.05. It was noted that typically the autocorrelation and cross-correlation of financial returns were not significant at any lag larger than one, so that the selection of L=3 aimed not at optimizing the procedure but rather at rendering it parameter-free. The procedure for the detection of structural breaks is described by Algorithm 1. If multiple structural breaks were found, we evaluated only those that seem relevant to the examined financial event.

#### 3.3.1. Financial Crisis in 2007–2008

For the dataset of daily returns on 37 stocks from the U.S. exchange market during the period 2004–2012, the beginning of the financial crisis in 2007–2008 was specified on 12 September 2008, when Lehmann Brothers collapsed. The network indices on the original time series and the VAR surrogates on sliding windows of size n=300 days and step s=150 days are shown in Figure 13.

Accordingly, the profile of the *p*-value of the test is shown in Figure 14 for the same dataset and sliding windows.

The *p*-value profiles indicated that four out of the six network indices, ave(k), ave(s), SD(s), and $ave(s^{+})$ (see Figure 14a,c,d,e), detected a structural break before the beginning (the collapse of the Lehmann Brothers on 12 September 2008), which was assigned by the non-rejection of $H_{0}$ in two consecutive windows followed by a rejection in the next window, indicating the emergence of nonlinear causalities. We observed in Figure 13 that each of the three average network indices was at a lower level than the respective empirical null distribution, indicating that the network density was smaller for the original data than for the linear surrogates. Indices SD(k) and $SD(s^{+})$ detected structural breaks long before the event, which was irrelevant to the studied breakout (see Figure 14b,f), and the same was true for the structural breaks after 2008, found by the indices ave(s), SD(s), and $ave(s^{+})$ (see Figure 14c–e). The latter may have indicated a reverse in the system’s behavior to a normal state after the violent movements that occurred after September 2008.

#### 3.3.2. Commodity Crises in the Second Half of 2014 and in April of 2020

In the second half of 2014, global commodity prices fell 38% between June 2014 and February 2015 as demand and supply conditions led to lower price expectations. Furthermore, in the beginning of 2020 and after the COVID-19 breakout, reduced global demand and problems in storage of oil, led the WTI (World Text Intermediate or Brent) future contract to close at a negative value. This situation remained until the middle of 2020 when there were signs of economic recovery. For the dataset of 8 daily future prices of commodities and MSCI of 7 countries regarding these 2 commodity crises, the computational setting was the same, and the profile of the *p*-value of the test and the 2 aforementioned breakpoints are shown in Figure 15.

Regarding the first commodity crisis in 2014, three network indices, ave(s), $ave(s^{+})$, and $SD(s^{+})$, detected a structural break just before the stated breakout of the crisis (see Figure 15c,e,f). Although, while ave(s) and $ave(s^{+})$ (see Figure 15c,e) signaled a change in the underlying mechanisms of the system by rejecting the null hypothesis regarding the absence of nonlinear causality, the $SD(s^{+})$ detected it in the opposite way. The index ave(k) (see Figure 15a) also found a structural break immediately following the first denoted event by indicating an emergence of nonlinear behavior. As far as the second commodity crisis was concerned, the indices ave(k) and ave(s) signaled a structural break before the breakout, while SD(s), $ave(s^{+})$ and $SD(s^{+})$ (see Figure 15d–f) detected it after the event. The network indices ave(s) and $ave(s^{+})$ indicated other breaks at the middle point of the examined financial events that were irrelevant for the current study. Finally, the index SD(k) (see Figure 15b) did not seem to be able to detect any of the denoted changes.

#### 3.3.3. Brexit Referendum in 2016

On 24 June 2016, Britains voted whether remain in the European Union. It was obvious that such an event would have an impact in U.K. economy. For this reason, we applied our proposed test on 17 stocks of the FTSE100 index during the period 2013–2019 and applied the same computational setting. The *p*-value profiles for the six network indices are shown in Figure 16.

We observed that ave(s) and $ave(s^{+})$ (see Figure 16c,e) detected an emergence of nonlinear effects two time windows before the referendum, while $SD(s^{+})$ (see Figure 16f) detected the same effect three time windows before, indicating a possible nervousness throughout the economy due to the unknown results of the Brexit vote. Moreover, $SD(s^{+})$ signaled a break immediately following the vote by not rejecting the null hypothesis after two consecutive rejections. This event could indicate a possible return to the normal system state. Regarding the rest of the network indices, ave(k) and SD(s) (see Figure 16a,d) stated a structural break right after the event while SD(k) (see Figure 16b) detected a break three time windows ahead of the referendum. All these indications were given by a rejection of the null hypothesis after two non-rejections in a row, thus signaling the appearance of nonlinearities in the system. Multiple breaks were also stated by all network indices, except from SD(k), before and after June 2016, but these were likely irrelevant to the examined financial event due to their time distance.

#### 3.3.4. COVID-19 Pandemic

On 30 January 2020, the World Health Organization (WHO) declared COVID-19 as a Public Health Emergency of International Concern. Thus, we used this date as a possible structural change on our system, which was comprised of 17 futures from government bonds and indices during the period 2019–2020. We applied the same computations using a sliding window of the size n=1000 and step s=500, since the data were hourly, and we derived the *p*-value profiles for the six network indices, shown in Figure 17.

We noticed that only ave(k) (see Figure 17a) detected a structural break before the WHO announcement, while SD(k), ave(s) and SD(s) (see Figure 17b–d) started to detect nonlinearities in the system two windows before the event, although the algorithm did not detect the early warning signal. The latter indices indicated a structural break immediately following the examined event, where the nonlinear causalities did not seem to dominate the system. This could be explained by the fact that on March 2020, the U.S. Federal Reserve had decided to support the economy by providing liquidity, hence the fear of a possible recession, and walked away. Finally, the $ave(s^{+})$ and $SD(s^{+})$ (see Figure 17e,f) did not detect any breaks.

## 4. Discussion

In this article, we proposed a novel method for detecting structural breaks in the underlying mechanism of a complex system that is observable through a multivariate time series where the breaks are caused by the emergence or the diminishing of nonlinear interactions. The procedure was based on a statistical test, where the null hypothesis stated that there is no nonlinear Granger causality relationships in the system. A resampling test was performed by generating linear VAR surrogate time series after the application of a specific (Gaussian) monotonous transformation on the marginal variables in order to remain consistent with the normality assumption of the model’s residuals. Using the nonlinear information-based Granger causality measure of PMIME, the causality network was formed. Different network indices were used as a test statistic and computed on the causality networks from the original and the surrogate time series.

The employed network indices focused more on the global structure of the system, rather than the local scale, as we wanted to explore the overall causality of the network. Moreover, the network indices were based on the existence of the causal connections (binary connections) as well as on their strength (weighted connections). We discovered that the latter indices performed better than the former ones, and the strength of a connection was more important than its existence, meaning that the change of the total strength of the network may not be related to a change of its structure. We also considered indices that relied on the existing connections. The purpose of using such indices was to detect small deviations among the null and original systems, as in large networks, these changes could have a low overall impact due to the high number of the possible connections.

First, a simulation study was performed to evaluate the proposed procedure of the statistical testing of nonlinear causality to detect structural breaks. The simulation scenarios included linear and nonlinear multivariate stochastic systems as well as multiple coupled chaotic maps. However, none of the network indices performed optimally and the best performance changed with the simulation system. In the current study, we used a statistical significance level α=0.05 for the hypothesis test, but in cases where structural change did not regard the emergence of vanishing of nonlinear causality but rather a substantial change in the strength of nonlinear causality, the α had to lower in value in order to detect the change. Certainly, cases where structural change regarding emergence, vanishing, or substantial change of linear causality would not be detected by the proposed procedure, but this seemed a less realistic scenario for real-world complex systems, such as in finance and biology. Furthermore, in cases where the nonlinearities were not quite as strong in the system, the user could consider using longer time series records to decrease the standard error of the test statistic, rendering the change statistically significant. Another reason to use longer time series was the inefficiency of PMIME on short time series to provide accurate estimates (as with any other nonlinear measure). However, the selection of longer time series windows was restricted by the violation of stationarity, as the generating mechanism of the observed time series was likely to change at relatively small time scale.

The proposed procedure was applied on rolling time windows of a time series record, signaling a possible structural break after the emergence or a diminishing of nonlinear causality effects. This event was triggered by a rejection of the null hypothesis after two consecutive non-rejections of it or by a non-rejection of the null hypothesis after two rejections, respectively, while different combinations of the number of non-rejections followed by rejections of the null hypothesis could be considered (and vice versa). We used the 2-to-1 scheme, as we reviewed transitions from a steady, normal state into an excited and possibly chaotic state (in the case of nonlinear emergence). For this reason, the length of the windows we used for the different was quite short.

The performance of the algorithm was evaluated on different financial products during multiple periods of crises over the last fifteen years. The indices that were based on the strength of the connections seemed to perform better than the indices that relied on the existence of the connection. The best performing network index was the ave(s) that represented the mean strength of the network, as in three out of four cases (the financial crisis in 2007–2008, the two commodity crises in the second half of 2014 and in April of 2020, and the Brexit Referendum in 2016) signaled an emergence of nonlinear effects a few time windows before the denoted breakout events. However, in the case of the COVID-19 pandemic, it indicated the non-dominance of nonlinearities in the system one time window after the pandemic was declared. This observation indicated that before a structural break in a complex system, the network of nonlinear causality appeared to lose or strengthen its density. Furthermore, the network index SD(s) indicated structural changes after all the major financial events had been examined. In order to determine which case was accurate, one should look at the tails of the test statistic. The trigger was not always provided by the same direction, as in the financial crisis in 2007–2008 and in the COVID-19 pandemic, there were no rejections of the null hypothesis after two consecutive rejections of it, while in the other events, the reverse was true (emergence of nonlinear effects). In any case, we noticed that the distribution of the strength of the network seemed to change before and after a violent disturbance of a complex system.

## Figures and Tables

**Figure 1 entropy-25-00370-f001:**
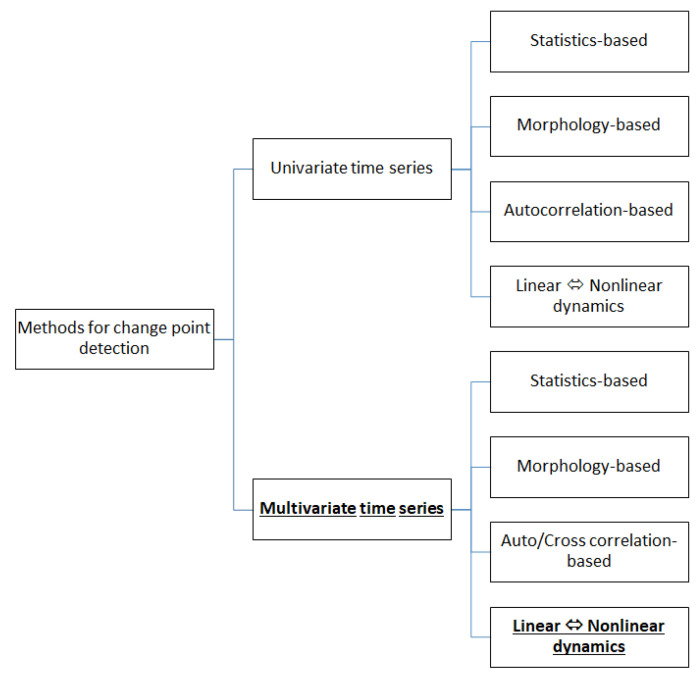
The main categories of the methods used for change-point detection in univariate and multivariate time series. Our proposed methodology detects structural breaks in multivariate time series by the emergence or the diminishing of nonlinear effects, thus it belongs to the subcategory in bold font.

**Figure 2 entropy-25-00370-f002:**
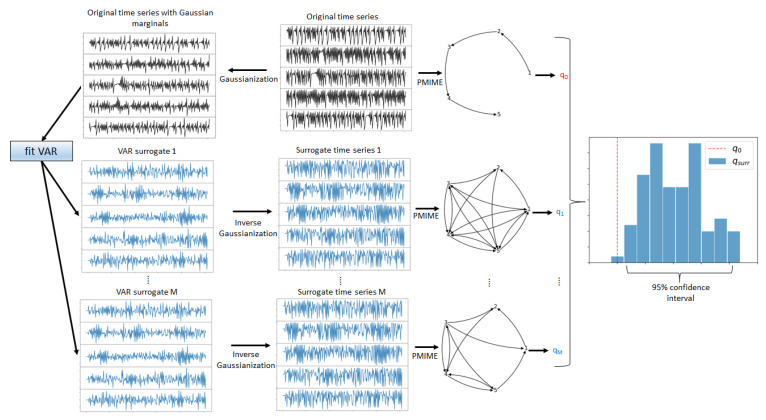
Graphical presentation of the test for nonlinear causality structure on a multivariate time series. For a *K*-dimensional time series (top panels, lines in black), the transform to Gaussian was applied to each univariate time series, and a VAR model was fitted. The VAR model was used to generate the *M* Gaussian *K*-dimensional time series (lower panels, lines in cyan), and each of them was transformed to the original marginal distribution. The causality networks for the original multivariate time series and the *M* surrogate multivariate time series were formed by the PMIME (the graphs on the right of the time series panels), and for each network the network index *q* was computed, so that $q_{0}$ was the test statistic on the original data and $q_{1},\ldots ,q_{M}$ for the surrogate data. The histogram on the right presents the empirical null distribution formed by $q_{1},\ldots ,q_{M}$, and the red vertical line stands for $q_{0}$. In this example, $q_{0}$ was at the tail of the distribution, denoted by the bounds of the 95% confidence interval under the $H_{0}$, suggesting the rejection of $H_{0}$ at the significance level α=0.05 and indicating the presence of significant nonlinear causality in the observed complex system.

**Figure 3 entropy-25-00370-f003:**
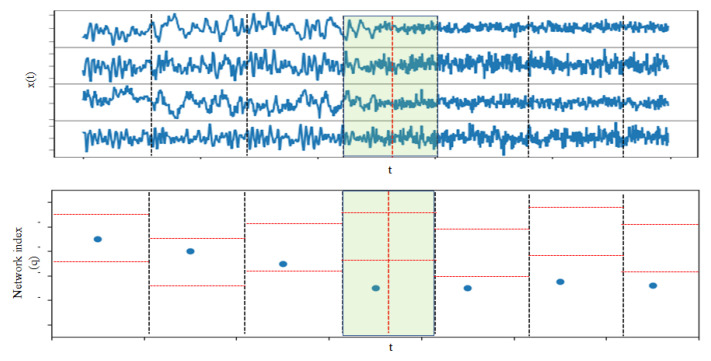
An example of the detection of structural break by applying the test for nonlinear causality on seven non-overlapping sliding windows of a multivariate time series, as shown in the top panel. In the lower panel, at each time window designated by vertical dashed black lines, the test statistic (a network index), and the corresponding α-significance bounds are denoted by a blue dot and red horizontal lines, respectively. The window of structural break is highlighted, and the vertical red line in the middle of the window stands for the estimated time of structural break, determined by the change of the test statistic moving out of the significance bounds.

**Figure 4 entropy-25-00370-f004:**
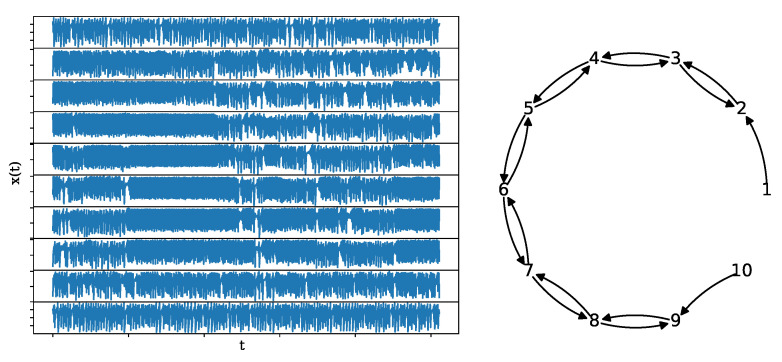
A realization of a system of 10 coupled Hénon maps (**left**) of length n=1024 along with its underlying network structure (**right**).

**Figure 5 entropy-25-00370-f005:**
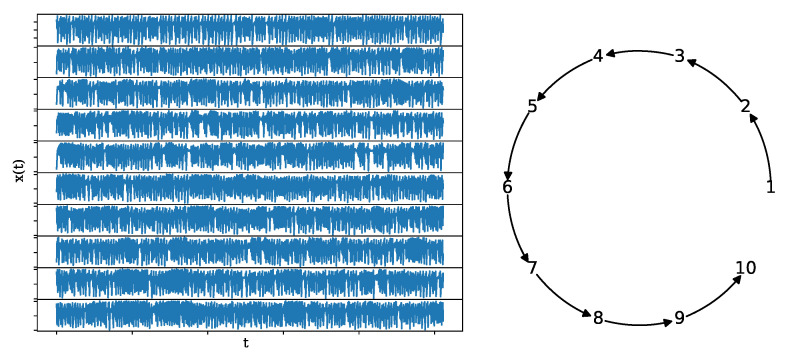
A realization of a causal Hénon map (**left**) of length n=1024, along with its underlying network structure (**right**).

**Figure 6 entropy-25-00370-f006:**
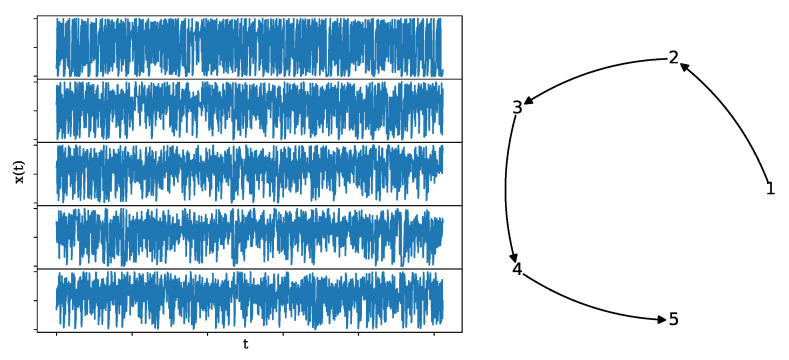
A realization of a causal logistic map (**left**) of length n=1024, along with its underlying network structure (**right**).

**Figure 7 entropy-25-00370-f007:**
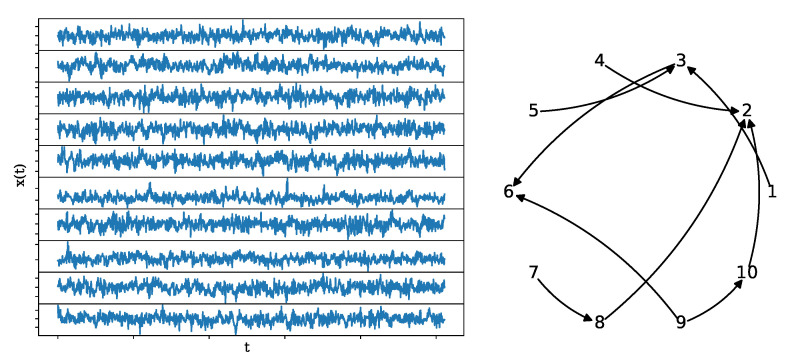
A realization of the NLVAR(4) system (**left**) of length n=1024, along with its underlying network structure (**right**).

**Figure 8 entropy-25-00370-f008:**
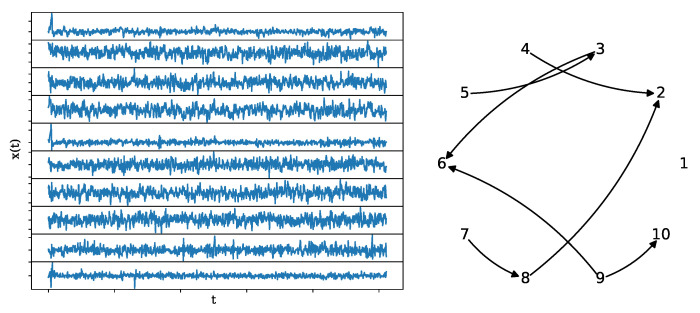
A realization of the VAR(4) system (**left**) of length n=1024, along with its underlying network structure (**right**).

**Figure 9 entropy-25-00370-f009:**
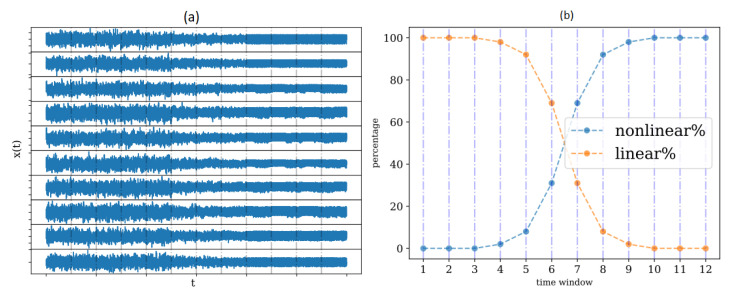
The 10-variate time series split into 12 non-overlapping time windows in (**a**), generated by the addition of the time series generated by the linear VAR(4) system and the time series generated by the nonlinear causal Hénon maps, each weighted by a percentage at each time windows shown in (**b**).

**Figure 10 entropy-25-00370-f010:**
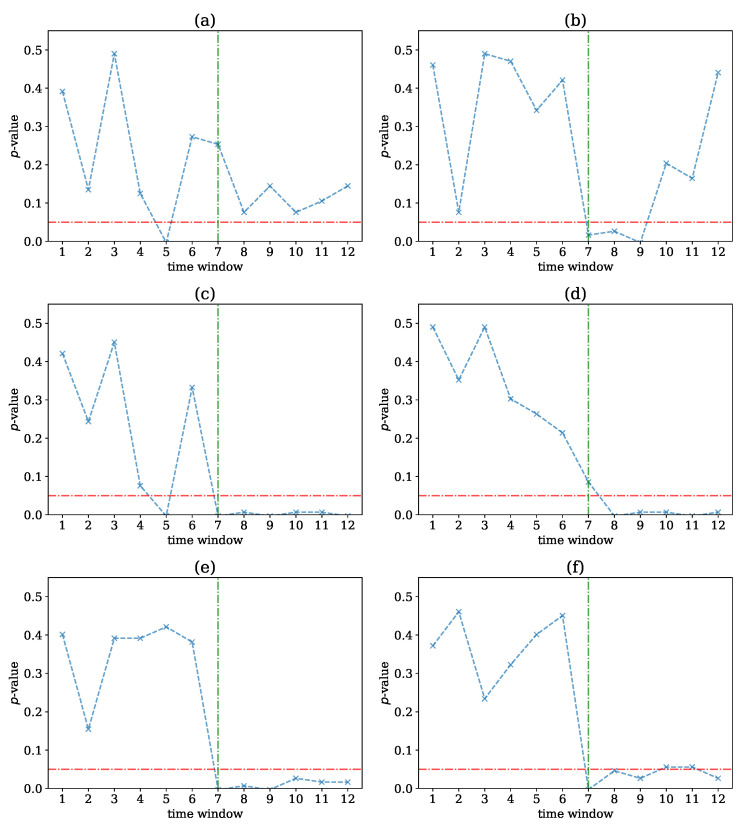
The *p*-value profile of the test for nonlinear causality across the 12 time windows of the exemplary time series, where the test statistic is the network index ave(k) in (**a**), SD(k) in (**b**), ave(s) in (**c**), SD(s) in (**d**), $ave(s^{+})$ in (**e**), and $SD(s^{+})$ in (**f**). The horizontal dotted red line denotes the significance bound of α=0.05, and the vertical dotted green line denotes the expected window of structural change occurrence.

**Figure 11 entropy-25-00370-f011:**
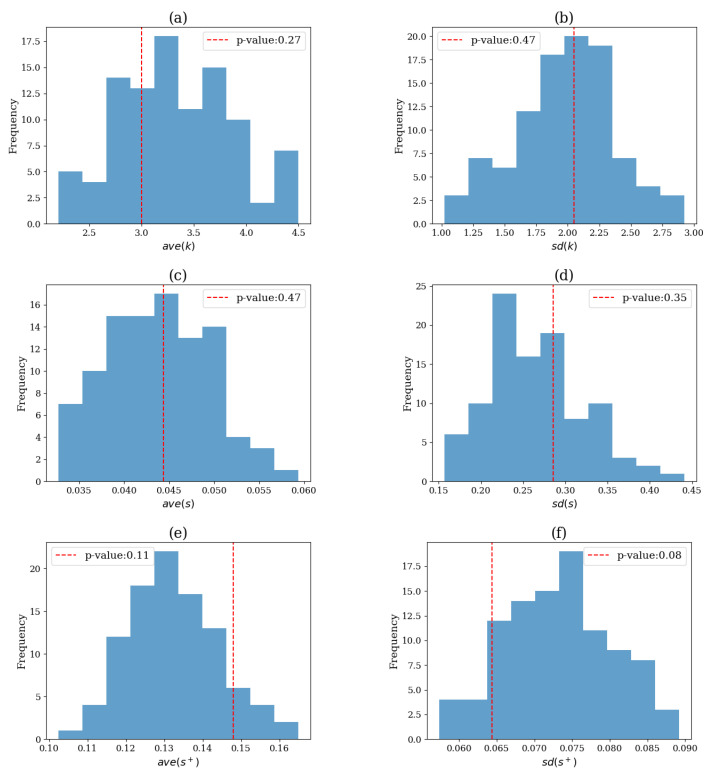
Histogram of the M=100 values of the test statistic on the surrogate data for the test for nonlinear causality on a single realization of the VAR(4) system. The red dashed vertical line denotes the test statistic value on the original time series and the legend displays the corresponding *p*-value. The test statistic is the network index ave(k) in (**a**), SD(k) in (**b**), ave(s) in (**c**), SD(s) in (**d**), $ave(s^{+})$ in (**e**) and $SD(s^{+})$ in (**f**).

**Figure 12 entropy-25-00370-f012:**
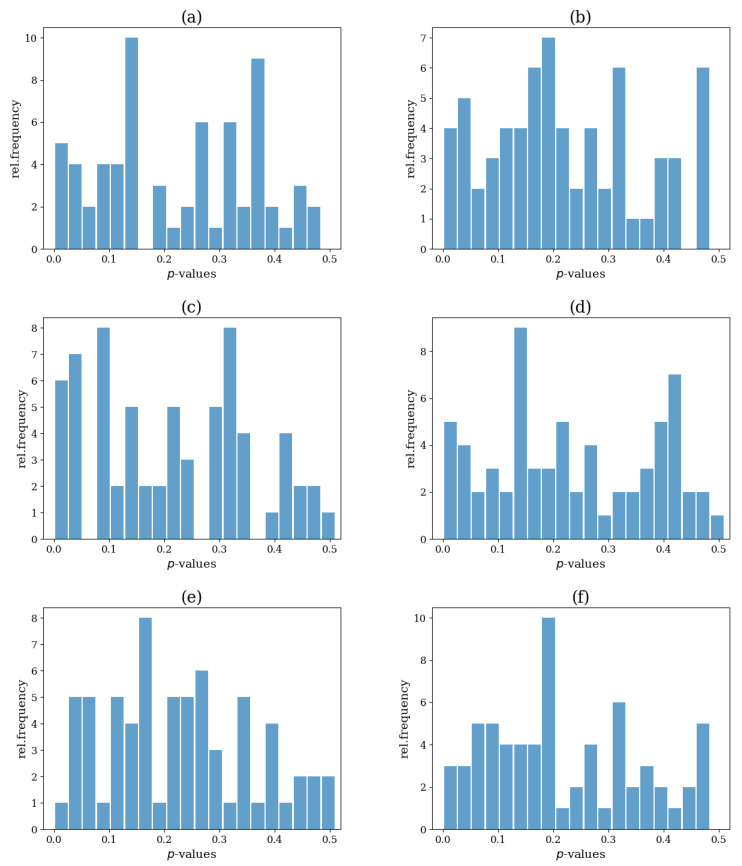
Histogram of the *p*-values for each test statistic after M=100 simulations for system VAR(4). The test statistic is the network index ave(k) in (**a**), SD(k) in (**b**), ave(s) in (**c**), SD(s) in (**d**), $ave(s^{+})$ in (**e**) and $SD(s^{+})$ in (**f**).

**Figure 13 entropy-25-00370-f013:**
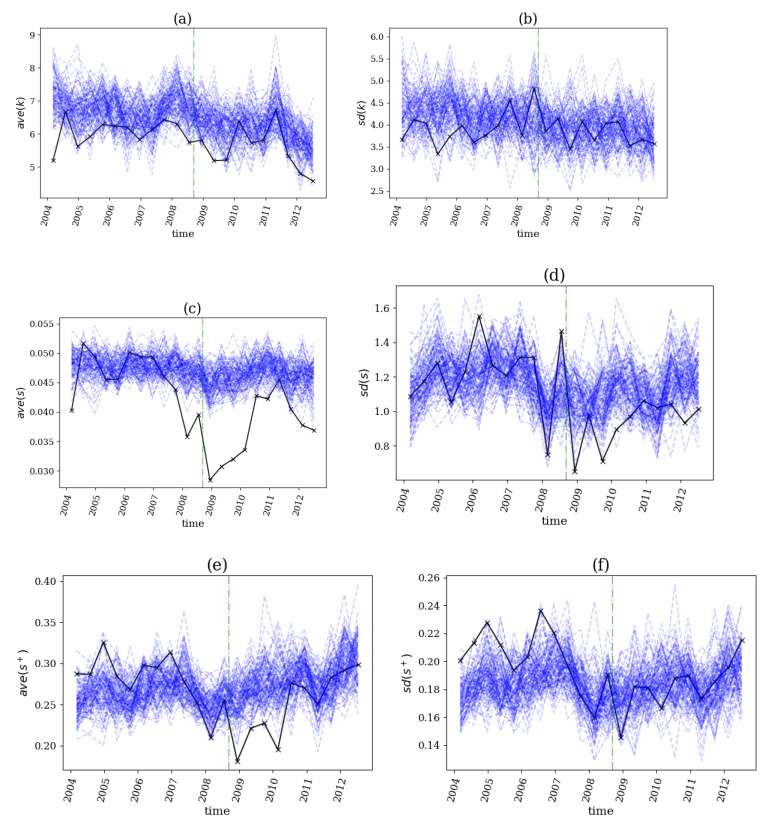
The network index profile on sliding windows of the dataset for the financial crisis in 2007–2008. At each panel, the black line is for the original time series and the 100 blue lines for the VAR surrogates, while the green vertical line denotes the start of the breakout. The network indices are ave(k) in (**a**), SD(k) in (**b**), ave(s) in (**c**), SD(s) in (**d**), $ave(s^{+})$ in (**e**) and $SD(s^{+})$ in (**f**).

**Figure 14 entropy-25-00370-f014:**
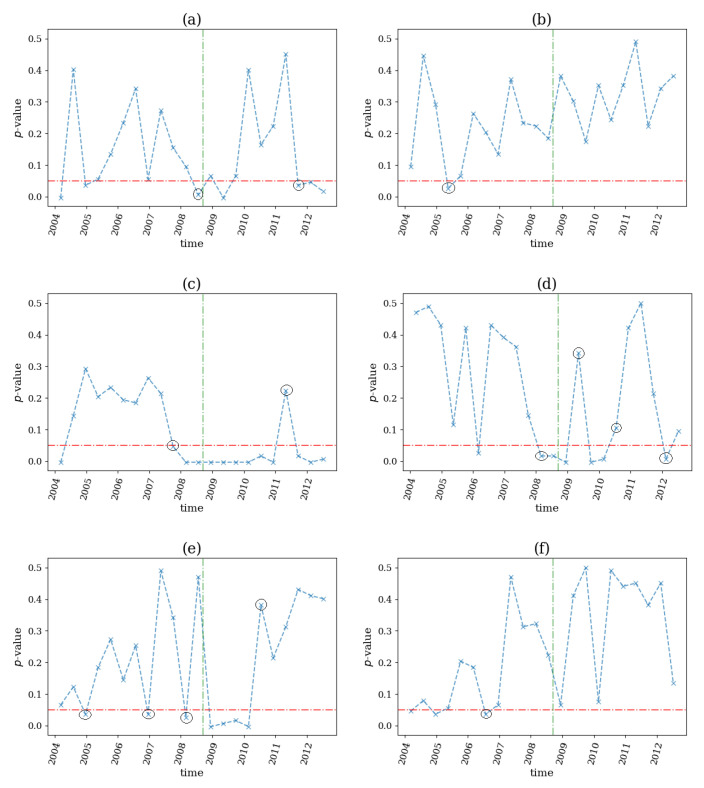
The *p*-value profile of the test for nonlinear causality on sliding windows of the dataset for the financial crisis in 2007–2008. The horizontal dotted red line denotes the significance bound of α=0.05, the green vertical line denotes the start of the breakout and the black circles denote the windows of structural break as identified by the proposed algorithm. The network indices used as statistics in the tests are ave(k) in (**a**), SD(k) in (**b**), ave(s) in (**c**), SD(s) in (**d**), $ave(s^{+})$ in (**e**) and $SD(s^{+})$ in (**f**).

**Figure 15 entropy-25-00370-f015:**
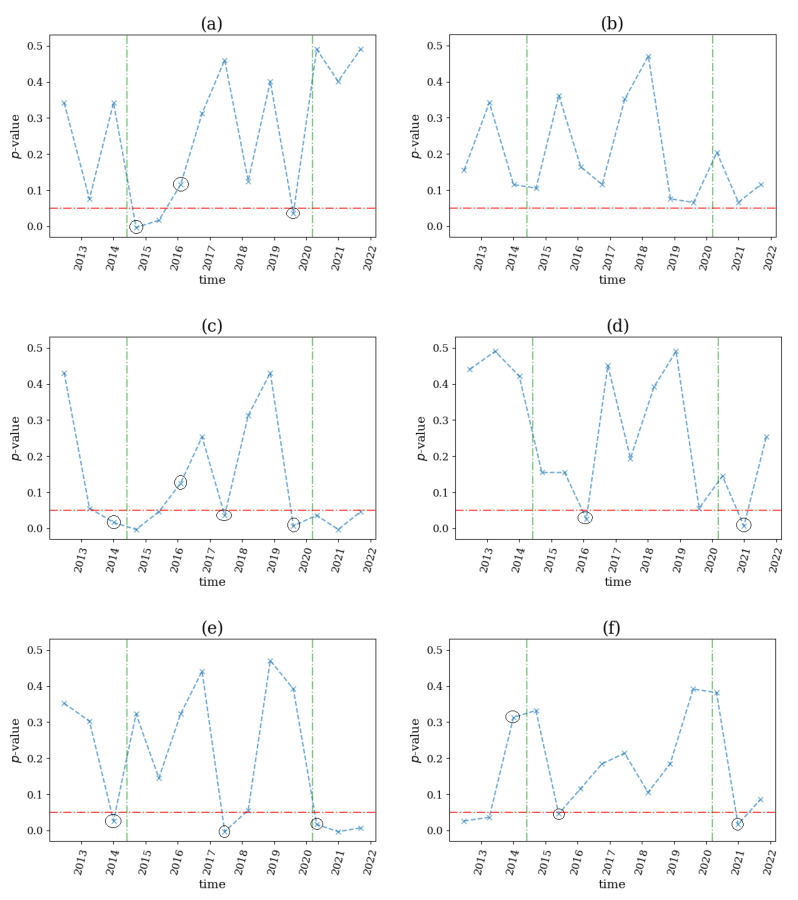
The *p*-value profile of the test for nonlinear causality across the time windows of the financial assets considered for the commodities crisis in the second half of 2014 and in April 2020, where the test statistic was the network index ave(k) in (**a**), SD(k) in (**b**), ave(s) in (**c**), SD(s) in (**d**), $ave(s^{+})$ in (**e**) and $SD(s^{+})$ in (**f**). The horizontal dotted red line denotes the significance bound of α=0.05, the vertical, green lines denote the breakout of the two commodity financial crises and the black circles denote the structural breaks as occurred by the proposed algorithm.

**Figure 16 entropy-25-00370-f016:**
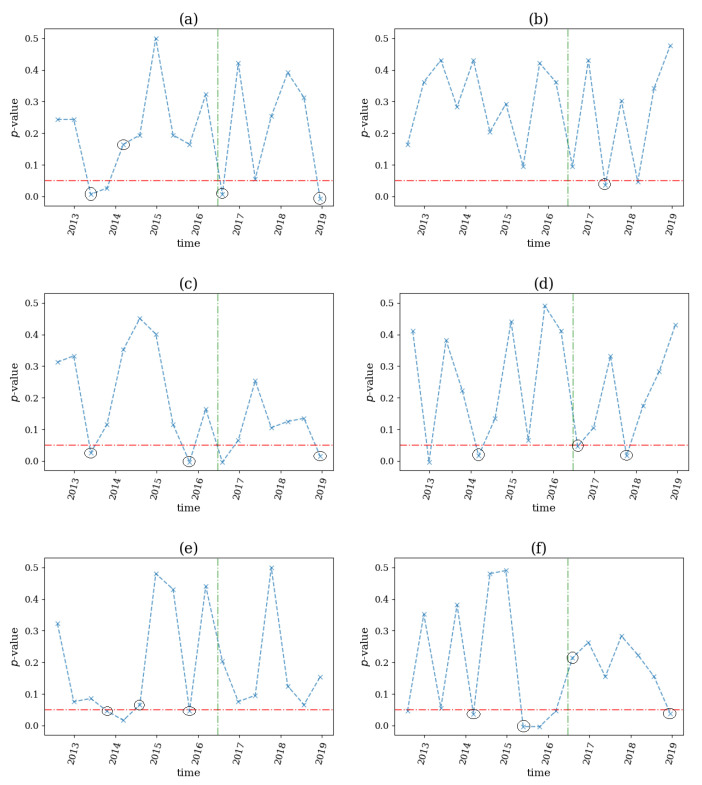
The *p*-value profile of the test for nonlinear causality across the time windows of the financial assets considered for the Brexit Referendum on 24 June 2016, where the test statistic is the network index ave(k) in (**a**), SD(k) in (**b**), ave(s) in (**c**), SD(s) in (**d**), $ave(s^{+})$ in (**e**) and $SD(s^{+})$ in (**f**). The horizontal dotted red line denotes the significance bound of α=0.05, the vertical, green line denotes the breakout of the financial crisis and the black circles denote the structural breaks as occurred by the proposed algorithm.

**Figure 17 entropy-25-00370-f017:**
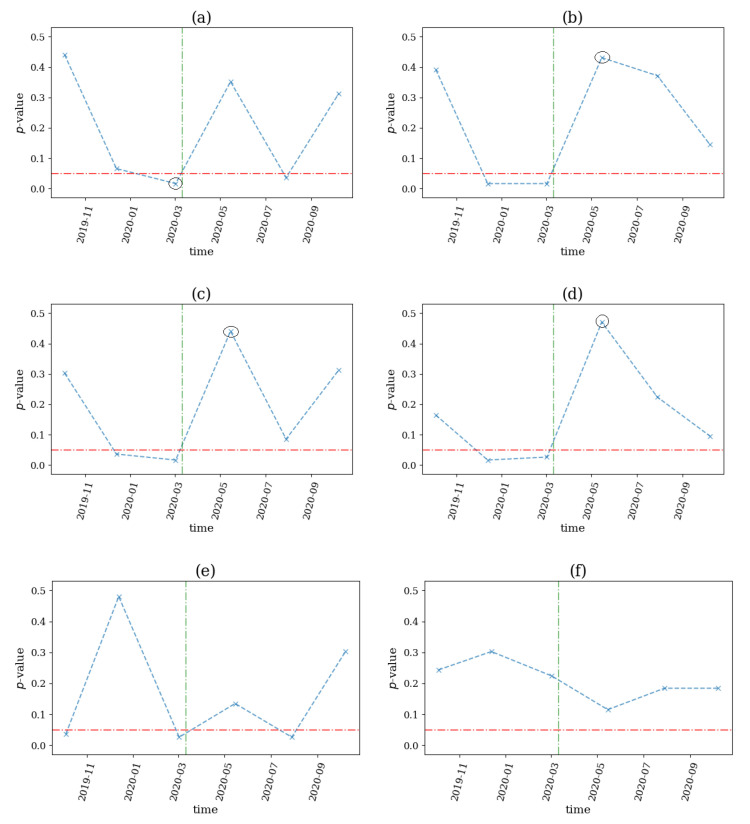
The *p*-value profile of the test for nonlinear causality across the time windows of the financial assets considered for the COVID-19 pandemic, where the test statistic is the network index ave(k) in (**a**), SD(k) in (**b**), ave(s) in (**c**), SD(s) in (**d**), $ave(s^{+})$ in (**e**) and $SD(s^{+})$ in (**f**). The horizontal dotted red line denotes the significance bound of α=0.05, the vertical, green line denotes the breakout of the financial crisis and the black circles denote the structural breaks as occurred by the proposed algorithm.

**Table 1 entropy-25-00370-t001:** Relative frequency of rejection of $H_{0}$ at the significance level α=0.05 from 100 realizations of the different systems (first column) and for the six network indices as test statistics (columns 2–7).

System	ave(*k*)	SD(*k*)	ave(*s*)	SD(*s*)	ave(*s*${}^{+}$)	SD(*s*${}^{+}$)
VAR(4)	0.13	0.13	0.19	0.13	0.09	0.09
NLVAR(4)	0.12	0.18	0.07	0.14	0.19	0.22
CausalHM	0.35	0.47	0.28	1.00	0.26	1.00
CoupledHM	0.27	0.24	0.45	0.52	0.41	0.08
CausalLM	0.04	0.16	1.00	1.00	1.00	0.28

## Data Availability

The artificial data used in this study and the Python code to generate them are available from the corresponding authors upon reasonable request. The real data used were obtained from https://finance.yahoo.com and http://www.msci.com.

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
