# Peer review of "Detecting Nonlinear Interactions in Complex Systems: Application in Financial Markets"

_entropy, 2023, doi:10.3390/e25020370_

Round 1

Reviewer 1 Report

the authors propose the detection of nonlinearities in financial data, in order to mine shift points. both simulated data and real data were used to illustrate the capabilities of the proposed methodology. the manuscript is well written, the mathematical aspect are sound, the results presented are encouraging. In my opinion this manuscript contains material that deserves publication. In the description of the literature, the authors may cite the effort which is spent in the last years to characterize high order properties of financial systems, like described in the paper Scagliarini et al, "Synergistic information transfer..." Entropy 2020, 22(9), 1000

Author Response

Please see attached Word document.

Reviewer 2 Report

The paper focuses on identifying significant changes in the underlying dynamics of a system, analyzing time series associated with the system.  To this end, the authors develop a specific statistical test. The test can be used to identify points where violations of series stationarity occur, which are attributed to a significant change in system dynamics by the appearance or disappearance of nonlinear terms. Subsequently, the authors apply the test developed to several economic events, the Great Recession of 2008, the impact of Brexit 2016 on the British stock market, the 2020 commodities crisis due to COVID. The article seems well-founded, entertaining and, potentially, very interesting.

However, some points have been difficult for me. For example, I cannot connect well how section 2.6 is used in the data (it would be desirable that sections 3.2.3 and 3.2.4 give additional details). Some  additional comments concerning Hénon maps are also missing in section 3.3.

The weakest point or that has offered me the greatest difficulties are the appearance of the black circles in figures 14, 15 and 16. This contrasts with figure 10 which seems easier to interpret. However, the black circles that seem to predict the  structural  breakdown in Figures 14, 15 and 16 appear both before and after the critical point and I do not understand by what criteria some points with black circles are indicated (sometimes they look like crossing lines sometimes local maximums) and in what way I can take these black circles as corroborations that a structural change has occurred or will occur.

Author Response

Please find attached Word document.
